# Scalable Generation of Nanovesicles from Human-Induced Pluripotent Stem Cells for Cardiac Repair

**DOI:** 10.3390/ijms232214334

**Published:** 2022-11-18

**Authors:** Jonathan Lozano, Alin Rai, Jarmon G. Lees, Haoyun Fang, Bethany Claridge, Shiang Y. Lim, David W. Greening

**Affiliations:** 1Baker Heart and Diabetes Institute, Melbourne, VIC 3004, Australia; 2Baker Department of Cardiovascular Research Translation and Implementation, La Trobe University, Melbourne, VIC 3086, Australia; 3Department of Microbiology, Anatomy, Physiology and Pharmacology, School of Agriculture, Biomedicine and Environment, La Trobe University, Melbourne, VIC 3086, Australia; 4Baker Department of Cardiometabolic Health, University of Melbourne, Melbourne, VIC 3010, Australia; 5Central Clinical School, Monash University, Melbourne, VIC 3004, Australia; 6O’Brien Institute Department, St Vincent’s Institute of Medical Research, Melbourne, VIC 3065, Australia; 7Department of Surgery and Medicine, University of Melbourne, Melbourne, VIC 3010, Australia; 8Department of Biochemistry and Chemistry, School of Agriculture, Biomedicine and Environment, La Trobe University, Melbourne, VIC 3086, Australia; 9National Heart Research Institute Singapore, National Heart Centre, Singapore 169609, Singapore; 10Drug Discovery Biology, Faculty of Pharmacy and Pharmaceutical Sciences, Monash University, Melbourne, VIC 3800, Australia

**Keywords:** nanovesicles, extracellular vesicles, tissue repair, proteomics, human iPSCs

## Abstract

Extracellular vesicles (EVs) from stem cells have shown significant therapeutic potential to repair injured cardiac tissues and regulate pathological fibrosis. However, scalable generation of stem cells and derived EVs for clinical utility remains a huge technical challenge. Here, we report a rapid size-based extrusion strategy to generate EV-like membranous nanovesicles (NVs) from easily sourced human iPSCs in large quantities (yield 900× natural EVs). NVs isolated using density-gradient separation (buoyant density 1.13 g/mL) are spherical in shape and morphologically intact and readily internalised by human cardiomyocytes, primary cardiac fibroblasts, and endothelial cells. NVs captured the dynamic proteome of parental cells and include pluripotency markers (LIN28A, OCT4) and regulators of cardiac repair processes, including tissue repair (GJA1, HSP20/27/70, HMGB1), wound healing (FLNA, MYH9, ACTC1, ILK), stress response/translation initiation (eIF2S1/S2/S3/B4), hypoxia response (HMOX2, HSP90, GNB1), and extracellular matrix organization (ITGA6, MFGE8, ITGB1). Functionally, NVs significantly promoted tubule formation of endothelial cells (angiogenesis) (*p* < 0.05) and survival of cardiomyocytes exposed to low oxygen conditions (hypoxia) (*p* < 0.0001), as well as attenuated TGF-β mediated activation of cardiac fibroblasts (*p* < 0.0001). Quantitative proteome profiling of target cell proteome following NV treatments revealed upregulation of angiogenic proteins (MFGE8, MYH10, VDAC2) in endothelial cells and pro-survival proteins (CNN2, THBS1, IGF2R) in cardiomyocytes. In contrast, NVs attenuated TGF-β-driven extracellular matrix remodelling capacity in cardiac fibroblasts (ACTN1, COL1A1/2/4A2/12A1, ITGA1/11, THBS1). This study presents a scalable approach to generating functional NVs for cardiac repair.

## 1. Introduction

The need for effective strategies to promote cardiac tissue protection and subsequent regeneration in the heart following a major cardiac event (i.e., myocardial injury, ischaemia) remains unmet [1,2,3,4,5,6]. Given the limited regenerative capacity of the human heart, stem cell-based therapies have emerged as a promising strategy for cardiac repair and restoring heart function [7,8]. However, clinical trials using cell-based therapy report a low engraftment rate and are associated with the risk of arrhythmia [9] and teratomas [8]. Importantly, the poor engraftment of stem cells post-transplantation suggests that secreted factors potentially mediated the observed cardiac repair [10,11].

Extracellular vesicles (EVs) have emerged as a key component of secreted factors, regulating cardiac repair [12,13,14,15,16]. EVs are 30–1000 nm-sized membranous vesicles released by cells, implicated in intercellular signalling in physiological and pathological processes [17]. EVs carry a variety of packaged proteins and nucleic acids that can be transferred to target cells to regulate functional response [18,19,20]. EVs from various stem cell sources have been studied for their capacity to regulate hallmark processes of cardioprotection and repair, including cardiomyocyte survival [21], revascularization [22], and regulating fibroblast activation [22] (scar tissue formation), as well as their regenerative properties in other tissues [23,24,25,26]. From induced pluripotent stem cell (iPSC) source EVs have demonstrated capacity to improve cell survival, angiogenesis, left-ventricular systolic function, while limiting remodelling and hypertrophy in an ischemia-reperfusion model in vitro and in vivo [14]. Further, iPSC-derived EVs reduced cell death and oxidative stress, while attenuating macrophage infiltration and preserving renal function in an in vivo acute injury ischemia-reperfusion model [27]. EVs from human iPSCs displayed anti-fibrotic function in hepatic stellate cells (in response to transforming growth factor-β (TGF-β)) [28] decreasing expression of α-smooth muscle actin (α-SMA), collagen Iα1 (COL1A1), and tissue inhibitor of metalloproteinases-1 (TIMP-1).

However, scalable generation of EVs (particularly from stem cells) remains an ongoing technical challenge that has limited their clinical utility [29,30,31]. Thus, in recent years, technologies to generate particles that mimic EVs (nanovesicles, NVs) directly from stem cells have garnered therapeutic interest for their high yield and reparative function [31,32,33,34,35,36,37]. For example, NVs have been generated in high quantity from mesenchymal stem cells (MSCs) with demonstrated myocardial protective effects in in vitro (pro-angiogenic, pro-survival) and in vivo (scar size reduction, preservation of cardiac function) settings [38]. However, MSCs are inherently difficult to isolate and maintain in large quantity [39,40]. This limitation can be addressed by employing pluripotent stem cells such as iPSCs that are easy to generate autologously from individual somatic cells (easily sourced from skin, hair, peripheral blood, and bodily fluids such as urine) [41,42,43], have unlimited proliferative capacity and can be maintained in culture long-term [44,45,46,47]. Therefore, iPSCs are proposed as a preferred alternative source for scalable generation of NVs. However, whether iPSC derived NVs have cardiac repair function remain unknown. Here, we reported a reproducible strategy to generate functional NVs from different human iPSCs efficiently and demonstrated their therapeutic potential as a functional surrogate for natural EVs for tissue repair.

## 2. Results

### 2.1. Validation of Human Induced Pluripotent Stem Cell Models

As a source of human iPSC donor cells for generating nanovesicles (NVs), we selected two well-established human iPSC lines, namely CL2 and CERA, reprogrammed from dermal fibroblasts [48,49] (Figure 1A). These iPSCs are defined by their expression of hallmark pluripotency markers, LIN28A, OCT4/POU5F1 and L1TD1, and display characteristic rounded morphology growing as island clusters [48,49,50,51]. In our study we verified that these iPSCs were highly proliferative (>99% Ki67 positive), have high cell viability (>94% propidium iodide negative), grow in compacted colonies and display morphological features characteristic of undifferentiated cells consistent with the original report [48,49] (Figure 1B–D, Appendix A). To ascertain their iPSC phenotype, we further analysed their cellular proteome using mass spectrometry (MS) and compared to HUVEC as a non-iPSC type (Figure 1E,F) (Appendix A). Pearson correlation analysis revealed that the two iPSC proteomes clustered together (>0.80) (Figure 1G). Moreover, K-means clustering identified 304 proteins that were significantly more abundant in iPSCs (FC > 1.5, *p* < 0.05) when compared to HUVEC proteome data; in contrast, 135 proteins were found in lower abundance (Figure 1I) (Appendix A). Highly abundant proteins in iPSCs include pluripotency markers (LIN28A, OCT4/POU5F1 and L1TD1) (Figure 1H) and are implicated in transcriptional and translational regulation (mRNA splicing, spliceosome), processes that are key regulators of iPSC formation [52] (Figure 1J,K). On the other hand, low abundant proteins included endothelial cell marker proteins, PECAM1, ANPEP and CD59, features of terminal differentiation [53,54] (Figure 1H). We thus verified the pluripotency identity of the two iPSC lines used in this study.

**Figure 1 ijms-23-14334-f001:**
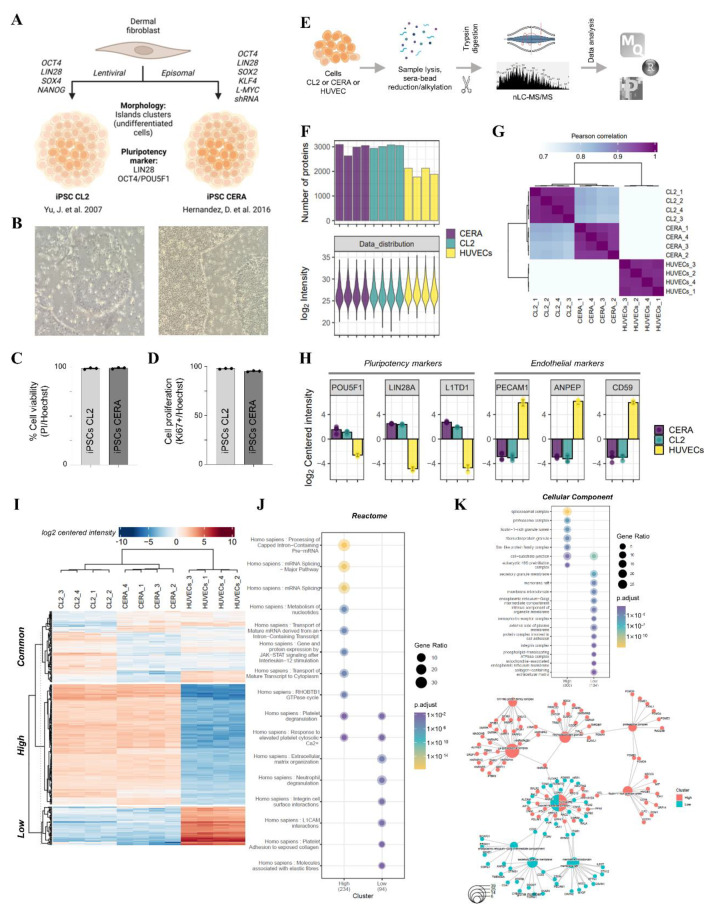
Validation of human-induced pluripotent stem cell lines. (**A**) CL2 and CERA iPSCs are reprogrammed from human dermal fibroblasts using lentiviral and episomal reprograming method, respectively. Pluripotency markers as identified in current study by cell-based proteomic profiling of each iPSC model (CL2/CERA). (**B**) Brightfield microscopy reveals derived iPSCs grow in an island-like pattern (20× magnification). (**C**) iPSC viability as assessed by IP/Hoechst staining (*n* = 3). (**D**) iPSC proliferation determined by Ki67 and Hoechst staining (*n* = 3). (**E**) Mass spectrometry-based proteomic profiling workflow, including sera-bead sample preparation with nano-liquid chromatography tandem mass spectrometry and data processing/informatics. (**F**) Protein identification and LFQ (normalised) intensity for iPSCs and HUVECs (*n* = 4). (**G**) Pearson correlation (R-score) of iPSCs (CL2, CERA) and HUVECs (*n* = 4). (**H**) Differential abundance (LFQ centred intensity, log2) of pluripotency and endothelial markers as determined by proteomic profiling (*n* = 4). (**I**) Hierarchical cluster analysis of iPSCs and HUVECs (ANOVA, *p* < 0.05, fold change (FC): 1.5), reveals three clusters of differential protein expression; *high* abundant iPSCs, *low* abundant iPSCs. Gene Ontology enrichment analysis using (**J**) Reactome and (**K**) Cellular Component for cluster *high* and *low* groups. Enrichment map highlighting proteins associated with each differential protein expression cluster (*high*/*low* in iPSCs) [48,49].

### 2.2. Generation and Characterization of Nanovesicles from iPSCs

We next generated NVs by serially extruding iPSCs through three filter membranes of decreasing pore size; this process termed serial extrusion [55] (Figure 2A). The extruded material was then subjected to density gradient separation (DGS), a gold standard for EV purification, whereby majority of proteins were enriched in fraction 5 with floatation density of ~1.13 g/mL, which is characteristic floatation density of natural EVs [56]. Protein distribution across density fractions resulted in 350 µg yield (~66% of total protein signal for both iPSC models) (Figure 2B). Further single particle tracking analysis highlighted striking enrichment/abundance of particles in fraction 5 (~1.6 × 10^15^ particles/mL) (Appendix A). To ascertain that this fraction contains NVs, we employed cryo-electron microscopy which revealed abundant membranous particles that are spherical in shape and morphologically intact, reminiscent of natural EV morphology (Figure 2C). We refer to these particles as CERA-NVs or CL2-NVs depending on their parental cell origin. Both cryo-electron microscopy and single nanoparticle tracking analysis revealed these NVs were ~110 nm in mean size (ranging ~50–300 nm) (Figure 2D,E). Single nanoparticle tracking analysis also showed that NVs diameter remained constant even after 10 freeze/thaw cycles (F/T, Appendix A).

We typically obtained ~98 µg of NVs from a 6-well plate iPSC culture at ~80% confluency (~1.7 µg of protein/cm^2^). We next compared the yield of NVs with natural EVs (purified using DGS as previously described [56]) released by the same number of both iPSCs. We found that NV yields were a striking 900-fold higher when compared to natural EV yield (Figure 2F). Thus, our data show we can generate NVs from iPSCs in large quantities, with consistent short processing time typically requiring less than 4 h.

### 2.3. Proteome Analysis Identify Stem Cell Markers and Tissue Repair Proteins in NVs

To gain insight into their proteome and potential for tissue repair, we subjected the NVs to MS analysis (Figure 2G). The NV proteome of ~2000 proteins, although smaller as compared to iPSC proteome of ~3000 proteins (Figure 2H,I) (Appendix A). Comparative analysis reveals 1146 proteins not encapsulated in NVs across both iPSC models (Figure 2H and Appendix A); Gene Ontology enrichment analysis (*p* < 0.0001) shows these proteins include nucleic acid/protein binding proteins (spliceosome/nuclear proteins) (Appendix A), consistent with previous report [55]. Contrastingly, NV proteomes displayed a dynamic protein abundance (spanning 5 orders based on LFQ intensities) (Figure 2J) with top ranked proteins similar to iPSC proteome, including iPSC marker LIN28A and L1TD1 (Figure 2J). For therapeutic interest, EVs as well as their engineered NV counterpart, should contain proteins critical for tissue repair [14,38,57]. Upon further inspection, top ranked proteins in NVs include hallmark proteins implicated in wound healing (FLNA, MYH9, ACTC1, ILK), hypoxia response (HMOX2, HSP90, GNB1), stress response/translation initiation (eIF2S1/S2/S3/B4), and extracellular matrix organization (ITGA6, MFGE8, ITGB1) (Figure 2J,K, Appendix A) [58,59,60,61,62,63,64,65,66,67]. Furthermore, we show that several of these proteins (HSP 20/27/70/90, GJA1, ITGA6/B1, HMGB1, ILK) have also been reported in natural EVs with tissue repair properties (Figure 2J,K) [1,2,3,4,5,68,69]. We identified three clusters of protein (K-means clustering) with differential abundance between NVs and iPSCs (high/similar/low) (Figure 2L). Gene Ontology analysis of NVs revealed enrichment of hallmark processes of iPSCs which include their translational machinery (ribosome, ribosome biogenesis, translation initiation, spliceosome, spliceosome complex) (Figure 2M and Appendix A). Other major class of cellular proteins extruded into NVs include mitochondrial proteins (*p* = 8.50 × 10^−71^). Proteins that are features of cell membrane include cell binding/focal adhesion (*p* = 3.09 × 10^−22^), cytoskeletal components (*p* = 2.33 × 10^−4^), and cell substrate interactions (*p* = 1.24 × 10^−21^) were also identified abundantly in NVs. Furthermore, KEGG pathway analysis revealed components of the ribosome pathway (*p* = 2.41 × 10^−24^) highly enriched in NVs, highlighting the potential of NVs to reprogram the translational landscape of target cells (Figure 2M).

### 2.4. Uptake of iPSC NVs by Different Cardiac Cells

Similar to natural EVs, engineered NVs should also possess the ability to interact with target cells, and potentially be internalized, to deliver cargo and execute functional response. Our NV proteome included proteins such as integrins (ITGA6/B1) and tetraspanins (ADAM10 and CD9), that are present on EV surface membrane [70] and mediate EV uptake [71]. We thus assessed whether NVs can be taken up by two cardiac cell types: cardiomyocytes (human iPSC-derived cardiomyocytes, CMs) and cardiac fibroblasts (primary human cardiac fibroblasts, hCFs). We labelled NVs with lipophilic tracer DiI and subjected labelled NVs to cushion-based centrifugation to separate unbound dye as previously described [56]. Fluorescence microscopy revealed that NVs are readily taken up by recipient cells within 2 h (Figure 3A,B). Labelled NVs appeared as punctate structures reminiscent of natural EVs that are taken up by target cells (Figure 3A,B, inset). Confocal microscopy analysis of recipient cells along z-axis revealed that NVs are internalised (Appendix A).

### 2.5. iPSC NVs Functionally Regulates Cell Survival, Angiogenesis, and Fibroblast Activation In Vitro

Next, given their proteome composition we investigated whether following uptake, these iPSC-derived NVs modulate function of target cells in context of cardiac repair. One of the most prevalent cardiovascular diseases includes ischaemic heart disease, in which the heart is starved of oxygen and nutrients due to an impaired blood supply [72]. Clinical intervention entails reperfusion to restore blood flow. However, reperfusion can paradoxically exacerbate cellular dysfunction and cell death, in a time-dependent manner [73]. Effective therapeutic strategy limiting this damage (termed ischaemia-reperfusion injury) should thus focus on maintaining CM survival and tissue re-vascularisation [6].

We first assessed whether NVs could promote survival of reoxygenated CMs exposed to low oxygen culture condition (4 h of hypoxia) (Figure 4A). A single dose of NV treatment significantly (*p* < 0.0001) reduced CM cell death when compared to vehicle treatment (Figure 4B,C). NVs from both iPSCs displayed similar degree of CM protection, with percentage of cell viability comparable to CMs cultured in normoxia condition. Similarly, NV-treated HUVECs displayed significantly (*p* < 0.05) enhanced organisation into tube-like structures on a Matrigel matrix, when compared to vehicle treatment (Figure 4D–F).

Cardiac fibrosis is characterized by elevated deposition of extracellular matrix proteins in the cardiac *interstitium* and contributes to systolic and diastolic dysfunction [74,75]. TGF-β is induced as an upstream regulator of cardiac fibrosis, promoting fibroblast activation (via upregulated expression of alpha (α)-smooth muscle actin (α-SMA)) that assemble into contractile fibres and increases ECM deposition [76]. We show that NVs from both iPSCs significantly (*p* < 0.0001) attenuated TGF-β mediated expression of α-SMA in human cardiac fibroblasts (hCF) (Figure 4G–I).

Collectively, our data show that NVs from both iPSCs protect cardiomyocytes and promote endothelial cell tubule formation against simulated ischaemia-reperfusion injury, and attenuate fibroblast activation induced by TGF-β.

### 2.6. NVs Reprogram Target Cell Proteome to Support Cardiac Reparative and Protective Phenotype

We next investigated whether the function of NVs can be further supported by changes in proteome of cells treated with NVs in Figure 4. MS quantified similar number of proteins in target cells following treatment with NVs or vehicle control. In all three functional assays, we found that NV-treated target cell proteomes displayed high correlation coefficient (>0.80) and clustered together, suggesting similar reprogramming capacity of NVs (Figure 5A). This further supports comparable level of NV-mediated CM survival, endothelial cell tube formation, and TGF-β-mediated α-SMA expression in hCF. We highlight MS-based validation of NV-attenuated protein expression of α-SMA in TGF-β-treated hCF (Figure 4I, shown by Western blotting) (Figure 5B).

For the CM survival assay, we identified 46 (clusters 1 and 3, Appendix A) of proteins that were significantly enriched, and 5 proteins were found in lower abundance in CM treated with NVs (cluster 2, Appendix A). These differentially abundant proteins include STAT4 [77], CNN2 [78], THBS1 [79,80], RHEB [81], IGF2R [82] (pro-cell survival proteins) (Figure 5C and Appendix A). In treated CMs, Gene Ontology enrichment analysis (Reactome) identified pathways associated with fatty acid metabolism (R-HSA-75876/75105/400451/8978868), a key process for ATP production in the heart [83,84,85] (Figure 5D). This enrichment was not observed when compared to untreated normoxia CMs suggesting NVs restore this metabolic balance (Table 1, Appendix A).

At a proteome level NV-treated HUVECs, compared to control treatment, showed significantly higher abundance of 33 proteins that included key angiogenesis regulators such as MFGE8 [86,87,88,89], MYH10 [90] as well as the cardiac function regulator VDAC2 [91] (Figure 5C and Appendix A). In NV-treated HUVECs, Gene Ontology analysis (Reactome) of these clusters identified a significant enrichment of processes and pathways associated with translation (*p* = 1.78 × 10^−6^), and glucose metabolism (*p* = 6.39 × 10^−4^) [92,93] (Figure 5D, Appendix A).

**Table 1 ijms-23-14334-t001:** iPSC NVs induce proteome reprogramming of target cells to support repair functions.

Assay	NV Treatment	ID	Reactome Enrichment Pathways	Proteins Mapped	*p*-Val	Description	References
Cardiomyocyte survival	Target cells increased expression following iPSC NV treatments	R-HSA-75105	Fatty acyl-CoA biosynthesis	ACSL3, ACSL4, HACD3	2.20 × 10^−4^	NVs restore fatty acid metabolism; alterations affect the remodelling and functional capacity of cardiac cell/tissue	[84,85]
		R-HSA-8978868	Fatty acid metabolism	ACSL3, ACSL4, HACD3, HSD17B4	2.40 × 10^−3^		
		R-HSA-191273	Cholesterol biosynthesis	FDFT1, MSMO1	3.36 × 10^−3^		
HUVECs tube formation (angiogenesis)	Target cells increased expression following iPSC NV treatments	R-HSA-70326	Glucose metabolism	PCK2, PFKL, SLC25A10, SLC25A11, SLC25A13	6.39 × 10^−4^	NVs restore basal glycolytic metabolism of endothelial cells	[92,93]
Cardiac fibroblast (TGFβ-mediated) activation	Target cells decreased expression following iPSC NV treatments	R-HSA-446353	Cell-extracellular matrix interactions	ACTN1, FBLIM1, FERMT2, FLNA, FLNC, RSU1VASP	9.45 × 10^−10^	NVs attenuate ECM remodelling, deposition, organization capabilities of hCFs	[75,96,97,98]
		R-HSA-1474244	Extracellular matrix organization	ACTN1, COL12A1, COL1A1, COL1A2, COL4A2, CTSD, ICAM1, ITGA1, ITGA11, ITGAV, P3H1, P3H3, PPIB, THBS1, TIMP2	3.54 × 10^−6^		
		R-HSA-216083	Integrin cell surface interactions	COL1A1, COL1A2, COL4A2, ICAM1, ITGA1, ITGA11, ITGAV, THBS1	8.39 × 10^−6^		
		R-HSA-1650814	Collagen biosynthesis and modifying enzymes	COL12A1, COL1A1, COL1A2, COL4A2, P3H1, P3H3, PPIB	1.58 × 10^−5^		
		R-HSA-3000171	Non-integrin membrane-ECM interactions	ACTN1, COL1A1, COL1A2, COL4A2, ITGAV, THBS1	7.61 × 10^−5^		
		R-HSA-1474290	Collagen formation	COL12A1, COL1A1, COL1A2, COL4A2, P3H1, P3H3, PPIB	1.07 × 10^−4^		
		R-HSA-446728	Cell junction organization	ACTN1, FBLIM1, FERMT2, FLNA, FLNC, RSU1, VASP	1.15 × 10^−4^		
		R-HSA-8948216	Collagen chain trimerization	COL12A1, COL1A1, COL1A2, COL4A2	1.94 × 10^−3^		

In NV-treated hCFs compared to control treatment, we identified 13 proteins with significantly lower abundance when compared to vehicle control treatment (Appendix A). These proteins are implicated in cell-extracellular matrix interactions (ACTN1) [94], collagen formation (COL1A1/2/4A2/12A1) [74,75], and extracellular matrix organization (ITGA1/11, THBS1) [95,96,97,98] (Figure 5C and Appendix A), which correlates with the functional attenuation of TGF-β-induced fibroblast activation (Figure 4G–I). Moreover, NV treatment significantly decreased the expression of CCN2, a key regulator of TGF-β pro-fibrotic effects and a direct inducer of collagen expression [76]. Gene Ontology analysis supported pathways and processes associated with NV treatment including focal adhesion (*p* = 4.02 × 10^−7^), ECM organisation (*p* = 7.62 × 10^−7^), and stress fiber (*p* = 4.46 × 10^−23^) (Figure 5D, Appendix A).

Therefore, we highlight the capacity of single dose treatment of NVs derived from iPSCs to reprogram target cell proteome to support cardiac reparative and protective phenotype.

## 3. Discussion

The therapeutic potential of extracellular vesicles (EVs) lies within their multimodal capacity to protect and repair damaged tissues [69,99,100,101,102,103,104,105,106] but challenges in their large-scale production are a bottleneck to their clinical utility [29,30]. In this study, we report a rapid size-based extrusion strategy to generate large quantities of EV-like membrane-limited nanovesicles (NVs) from human iPSCs of therapeutic potential. While EV isolation takes approximately 22 h from conditioned media collection, and EV purification [107,108], NVs can be obtained from the same number of donor cells within 4 h. Importantly, following density purification, the fraction containing the highest protein and particle yield resulted in a 900-fold higher protein amount of NVs compared to purified EVs. We also showed that the diameter/size of NVs is not affected by consecutive freeze–thaw cycles. Collectively, our study highlights scalability, ease-of-use, high-throughput and shorter production time, selective use of donor cells and economical—all important logistical considerations for EV-based therapies [30,41].

Additional considerations for therapeutic utility include NVs ability to influence target cell function, carry bioactive cargo of interest, and induce a desired functional response [109]. Functionally, we show that these NVs can be taken up by various cell types (cardiomyocytes, cardiac fibroblasts, and endothelial cells). We chose two iPSC cells as a source of NVs because they contain proteins that influence cardiomyocyte survival, regulate fibroblast activation, and promote angiogenesis, several of which were packaged into NVs during extrusion. These include proteins implicated in wound healing (ACTN1, FLNA, MYH9, ACTC1, ILK), stress response/translation initiation (eIF2S1/S2/S3/B4), hypoxia response (HMOX2, HSP90, GNB1), and extracellular matrix organization (ITGA6, MFGE8, ITGB1). Consistent to functional pathway enrichment in NV cargo, NVs also elicit a heterogeneous response in target cells along with their proteome reprogramming supporting the functional response (Figure 3, Figure 4 and Figure 5). Contrastingly, components not enriched in NVs compared to parental cells included nucleic acid/protein binding proteins (spliceosome/nuclear proteins) consistent with a previous report; why there is this difference in the NV composition warrants further investigation [55].

In fact, several of these proteins such as HSP20/27/70/90, GJA1, ITGA6/B1, HMGB1, and ILK that have been shown to be packaged in natural EVs from stem cells and dictate reparative function (angiogenesis, proliferation, migration, immune modulation, and cell survival) [1,2,3,4,5,68,69]. Naturally secreted EVs versus NVs potentially have different molecular composition; EV biogenesis includes active cargo sorting mechanisms [71,110,111] whereas NV generation involves random cargo sampling from parental cells [55]. Despite these potential cargo differences, our iPSC NVs were able to recapitulate cardiac repair functions attributed to naturally produced EVs from stem cells in the field [12,13,14,15,16,112,113]. Our functional data showed that after hypoxia, NVs upregulate cell survival proteins (STAT4 [77], CNN2 [78], THBS1 [79,80], RHEB [81], IGF2R [82]) and promote cardiomyocyte survival. In the same context, NVs also supported endothelial tube formation and elevated expression of proteins associated with angiogenesis (MFGE8 [86,87,88,89], MYH10 [90]). Furthermore, NVs attenuated TGF-β mediated activation of human cardiac fibroblasts and its consequential expression of pro-fibrotic proteins (ACTN1 [94], CCN2 [76], COL1A1/2/4A2/12A1 [74,75], ITGA1/11 [95], THBS1 79,97,98]). Altogether, these results suggest that NVs regulate distinct and key processes of tissue repair that could be applied to the heart in the context of hypoxia/reoxygenation.

Current treatment following MI is the restoration of blood flow in the heart (reperfusion). If reperfusion occurs promptly (less than 2 h) after the infarct there is very limited damage [114]. Often, patients are intervened more than 2 h after the MI, which attributes increased tissue damage and heart failure [115]. Following reperfusion, treatments only prevent further ischemic events and have not shown any reparative effects to the infarct zone [116]. Indeed, preservation of tissue and cellular function is imperative to maintain functionality of the heart, with multiple signalling pathways involved that influence survival through stress fibre formation and contractile proteins, anti-apoptotic and cardiomyocyte survival, and promotion of blood vessel growth and development [6]. NV treatment during reperfusion represents a potential combination therapy to induce repair by promoting angiogenesis, cardiomyocyte survival and attenuation of extracellular matrix deposition. Such emerging strategies using NVs could further impact cardiac senescence and aging, delivering stemness factors (including PIM1 [117] or nucleostemin/GLN3 [118]) to influence impairment of myocardial biology and tissue repair following injury. Whether iPSC NVs could deliver such regulatory cargo (GLN3 [118], interestingly present in both iPSC source NVs) that may impact expression levels of markers of stemness and multipotency for reparative function remains to be determined.

Knowledge of the functional cargo that therapeutic EVs deliver could be exploited to engineer loaded NVs via robust incorporation (either during extrusion, i.e., active loading or via sonication/incubation, i.e., passive loading) [119]. Moreover, our extrusion process is also amenable to introducing targeting moieties either as lipid-conjugated or lipid binding protein-conjugated targeting peptides, and hence warrants further investigation [120,121,122,123,124,125]. Some of the key advantages of NVs include their low immunogenicity, intrinsic cell targeting properties, and enhanced stability in circulation which make them attractive in targeted drug therapy [30,126]. Future studies investigating their administration pathways, pharmaco-kinetics/-dynamics, dosing, biodistribution, and clearance, and therapeutic efficacy in pre-clinical models is a prerequisite [30,109,126,127].

Similar to EV-based therapeutics, the choice of donor cells for NV generation is critical. Indeed, EVs from different cell origins contain diverse cargo [110,111,128] (including surface membrane composition [70]) and exert distinct functions [129,130]. Although MSCs are extensively utilised, their isolation remains difficult, time consuming and/or invasive (liposuction, surgical resection, umbilical cord, or bone marrow aspiration) [131,132,133]. In contrast, iPSCs, which can be generated from skin (fibroblasts and keratinocytes) [134,135], extraembryonic tissues [136], peripheral blood [137], cord blood [136], and urine [42,43], have shown similar reparative effects [14,15]. Moreover, iPSCs maintain pluripotency even after >50 passages [44,45,46,47] important for scalability. In this study, as a proof of principle we show that NVs generated from two types of iPSCs (lentiviral [49] and episomal [50]) also display reparative function. We show that although highly similar in their cargo and target cell effect, differences in cell and NV proteome from different iPSC donors may contribute to such observations in cell uptake/internalisation and tissue reparative function. Whether this further impacts the membrane composition of NVs to alter capacity of cell targeting remains to be determined.

## 4. Materials and Methods

### 4.1. Cell Culture and Differentiation

#### 4.1.1. Human Induced Pluripotent Stem Cells (hiPSCs)

Human iPS-Foreskin-2 (CL2) cell line [49], kindly provided by James A. Thomson (University of Wisconsin), and CERA007c6 (CERA) [48] iPSC line were maintained on vitronectin-coated plates in TeSR-E8 medium (Stem Cell Technologies, VA, USA) according to the manufacturer’s protocol. Briefly, cells were cultured until confluent (7 days), enzyme free reagent (ReLeSR, Stem Cell Technologies) was used to detach cells and cell aggregates were re-seeded into new vitronectin pre-coated plates in a wash-free manner. Media was replaced every 2-3 days. Brightfield images were obtained using an inverted microscope (Olympus IX71, Tokyo, Japan) at 40× magnification.

#### 4.1.2. Human iPSCs Cardiomyocytes Differentiation and Culture

Cardiomyocytes (CMs) were derived from human iPSCs as previously described with modifications [138,139]. Briefly, human iPSCs were seeded onto human embryonic stem cell qualified-Matrigel (Corning) coated plates at a density of 1.25 × 10^5^ cells/cm^2^ in TeSR-E8 medium supplemented with 10 μM Y-27632 (Abcam, Waltham, MA, USA). After 48 h (cells ~90% confluent) (day 0), medium was replaced with RPMI-1640 basal medium (Thermo Fisher Scientific, Waltham, MA, USA) containing B-27 without insulin supplement (Thermo Fisher Scientific), growth factor reduced Matrigel (1:60 dilution) and 10 μM (for CL2 iPSCs) or 6 μM (CERA iPSCs) CHIR99021 (Cayman Chemical). At day 1, medium was replaced with RPMI 1640 basal medium containing B-27 without insulin supplement. At day 2, medium was changed to RPMI 1640 basal medium containing B-27 without insulin supplement and 5 μM IWP-2 (Sigma-Aldrich, Burlington, MA, USA) for 72 h. From day 5, cells were cultured in RPMI 1640 basal medium containing B-27 supplement (Thermo Fisher Scientific) and 200 μg/mL L-ascorbic acid 2-phosphate sesquimagnesium salt hydrate (Sigma-Aldrich) (cardiomyocyte medium, CMm), with CMm changed every 2–3 days. At day 12, differentiated cardiomyocytes were dissociated into single cells and split 1:4 onto human embryonic stem cell qualified-Matrigel coated plates in DMEM/F-12 GlutaMAX medium supplemented with 20% FBS (Sigma-Aldrich), 0.1 mM 2-mercaptoethanol, 0.1 mM nonessential amino acids, 50 U/mL penicillin/streptomycin and 10 μM Y-27632. At day 13, medium was changed to CMm. From days 14–19, CMs were enriched to >95% cardiac troponin T positive cells by culturing in glucose-free DMEM medium (Thermo Fisher Scientific) supplemented with 4 mM lactate (Sigma-Aldrich).

#### 4.1.3. Human Umbilical Vein Endothelial Cells (HUVEC) Culture

HUVECs (RFP expressing human umbilical vein endothelial cells, Angio-Proteomie cAP-0001) were cultured in EGM™-2 Endothelial Cell Growth Medium-2 BulletKit™ (LONZA CC-3156 and CC-4176 supplements) prepared as detailed by supplier. Medium was changed every third day, split when confluent, and reseeded at 1.0 × 10^4^ cells/cm^2^. Cells were harvested using 0.05% trypsin-EDTA (Gibco, Waltham, MA, USA) and plated as required on 1% gelatin (bovine skin Type B, Sigma-Aldrich, 9000-70-8) pre-coated plates.

#### 4.1.4. Human Cardiac Fibroblasts (hCF) Culture

hCF, Atrium, Adult, 306AK-05a) were cultured in 50% DMEM/F12, 40% cardiac fibroblast growth medium (Cell Applications, 316–500), 10% fetal bovine serum (FBS), split when confluent, and re-seeded in a 1:3 ratio. Cells were harvested using 0.05% trypsin-EDTA (Gibco) and plated (pre-coated with 1% gelatin) as required.

### 4.2. Cell Proliferation Assay 

Cell proliferation was performed by immunostaining with Ki67 antibody. Briefly, cells cultured on coverslips were fixed in 10% neutral buffered formalin and permeabilised with 0.2% triton X-100. After blocking with serum-free blocking solution (Thermo Fisher Scientific) for 10 min, cells were incubated with primary antibody against Ki67 (0.29 µg/mL, rabbit monoclonal IgG; Abcam) antibody and counterstained with 4′6-diamidino-2-phenylindole (DAPI, 1 µg/mL; Thermo Fisher Scientific) for nuclear staining. Images were taken using BX-61 Olympus fluorescence microscope (Tokyo, Japan). The number of proliferative cells (Ki67 positive) was counted from three random fields and expressed as a percentage over total number of cells (DAPI positive).

### 4.3. iPSC Nanovesicle Generation and Purification

Generation of nanovesicles (NVs) was performed as described [140] with modifications. Briefly, adherent iPSCs (CL2 and CERA) were individually harvested (following PBS wash) using a solution of EDTA (10 mM) (3 × 3 min round incubation) and cell suspension spun at 500× *g* for 5 min. The pellet was resuspended in PBS and the cell suspension sequentially extruded through 10, 5, and 1 µm polycarbonate membranes (19 mm; Advanti Polar lipids, 610010) (13 times across each filter, Whatman). The extruded NVs were subsequently purified using 10% OptiPrep™ (Stemcell Technologies) density cushion (step gradient formed by overlaying extruded sample on 10% and 50% iodixanol) and centrifuged at 100,000× *g* for 2 h at 4 °C. Seven equal fractions were obtained, diluted in PBS (to 1.5 mL), centrifuged at 100,000× *g* for 2 h at 4 °C (TLA-55 rotor; Optima MAX-MP ultracentrifuge) and resuspended in PBS and stored at −80 °C until further use. The yield (protein) and density of each fraction was determined as described [56].

For comparison, natural extracellular vesicles (EVs) from each iPSC line were obtained as described previously [56], based on isopycnic (iodixanol density-based) ultracentrifugation [108].

### 4.4. Cryo-Electron Microscopy

Cryo-electron microscopy (Tecnai G2 F30) on NVs was performed as described [108,141]. Briefly, NVs from each iPSC model (~1 μg protein, non-frozen samples prepared within 2 days of analysis) were transferred onto glow-discharged C-flat holey carbon grids (ProSciTech Pty Ltd., Kirwan, Australia). Excess liquid was blotted and grids were plunge-frozen in liquid ethane. Grids were mounted in a Gatan cryoholder (Gatan, Inc., Warrendale, PA, USA) in liquid nitrogen. Images were acquired at 300 kV using a Tecnai G2 F30 (FEI, Eidhoven, The Netherlands) in low dose mode. Size distribution of particles was calculated for the 12 fields of view (~300 different vesicles for each preparation).

### 4.5. Nanoparticle Tracking Analysis

Particle concentration and size was determined using nanoparticle tracking analysis (NTA, ZetaView, Particle Metrix, PMX-120; 405 nm laser diode) for all density fractions (volume normalized) [108]. Samples were prepared in 1 mL of PBS (14190-144, Thermo Fisher Scientific) and particle diluent analyzed in experimental triplicate; 11 positions were captured with the following parameters: camera sensitivity: 80, min area: 5, max area: 1000, brightness: 30, min trace length: 15, temperature: 25 °C. Calibration beads (Nano FCM, S16M-Exo) were used for instrument setup. Capture was performed at medium video setting, corresponding to 30 frames per position. ZetaView software 8.5.10 was used to analyse acquired data. For freeze–thaw analyses, NVs were placed on dry-ice, thawed and this cycle repeated for 1, 5, and 10× before NTA analysis.

### 4.6. NV Recipient Cell Uptake

Human CMs and CFs were seeded and cultured on gelatin coated glass bottom chamber µ-slides (8 well, Ibidi) to ~80% confluency. The medium was supplemented with DiI-labelled NVs (6 μg, concentration of 30 μg/mL) or PBS vehicle control at 37 °C for 2 h to allow uptake. For NV staining, NV was incubated with fluorescent dye DiI (Vybrant™ DiI Cell-Labeling Solution, 1:200 dilution, Invitrogen, Waltham, MA, USA, V22885) at 1 μM concentration for 15 min at 37 °C as described [142]. Briefly, labelled NV and PBS (DiI) were centrifuged at 20,000× *g* for 50 min on a 10% OptiPrep™ cushion, washed with PBS and resuspended in 50 µL of PBS. Following uptake, cells were washed twice in PBS and fixed using 4% formaldehyde for 5 min. Nuclei were stained for 7 min with Hoechst 33342 stain (Thermo Fisher Scientific) (10 μg/mL) prior to imaging by Nikon A1R confocal microscope equipped with resonant scanner, using a 20× WI (1.2 NA); (Nikon, Tokyo, Japan). Images were sequentially acquired. The XY image resolution was 1024 × 1024 at 0.033 FPS, 4× averaging, 2.4 dwell time. 3D images were taken by Z-stack of approximately 15 μm, 25 steps, at a resolution of 1024 × 1024, 8× averaging 2.4 dwell time. NS studio was used to render images.

### 4.7. In Vitro Model of Hypoxia/Reoxygenation

Human cardiomyocytes (CM) and human endothelial cells (HUVECs) at ~80% confluency were placed inside a hypoxia incubator chamber, and purged with nitrogen gas for 5 min The cell chamber was placed inside an incubator at 37 °C for either 4 h (for CMs) or 24 h (for HUVECs). After the hypoxia period, cell conditioned medium was replaced with respective culture media containing NVs (30 µg/mL) or PBS and cultured in 37 °C humidified 5% CO_2_ incubator for 24 h (for CMs) or 6 h (for HUVECs) to simulate reoxygenation.

### 4.8. iPSCs and Cardiomyocyte Survival Assay

For iPSC characterisation and CM survival following hypoxia/reoxygenation, cell viability was determined by 3 µg/mL propidium iodide (PI, Thermo Fisher Scientific, Waltham, MA, USA) and 5 µg/mL Hoechst 33258 (Sigma-Aldrich, St. Louis, MO, USA) dual staining (30 min). Images were captured at 200× magnification with an inverted microscope (Olympus IX71). The number of dead cells (PI positive) were counted from three random fields for each replicate for a total of 4 replicates each and expressed as a percentage of total cells (Hoechst 33258 positive).

### 4.9. Tube Formation Assay

Following 24 h of hypoxia, HUVECs were plated either into 96-well µ-plates (ibidi) for microscopy or flat-bottom 96-well plates for MS. Cells were treated with either NVs (single dose, 30 µg/mL) or vehicle control (PBS), and incubated in normoxia conditions (37°C, 5% CO_2_, humidified incubator) for 6 h. Cells were washed with PBS and bright field images obtained using an inverted microscope (Inverted Confocal Nikon A1r Plus NIR) at 200x magnification. Manual identification of tubes was performed followed by ImageJ quantitation. Results are shown as average number of tubules per condition.

### 4.10. Fibroblast TGF-β-Mediated Activation Assay

hCF cells were seeded at confluency in 24-well plate pre-coated with 1% gelatin and incubated for 24 h at 37 °C, 5% CO_2_ to allow the cells to attach. Cells were serum-starved for 24 h (to obtain basal levels of α-SMA; activation) prior to treatment with 5 ng/mL of TGF-β (or non-treatment control) for 24 h (conferring hCF activation/α-SMA expression), followed by treatment with NVs (30 µg/mL; 9 µg) or vehicle control (PBS) for 24 h at 37 °C, 5% CO_2_. Medium was removed, followed by 3 washes with PBS, and cells were immediately lysed in-well with cell lysis buffer (1% SDS, 50 mM HEPES pH 8) on ice for 5 min, heat treated at 95 °C for 5 min prior to protein quantification (microBCA™ Protein Assay Kit, Thermo Scientific, 23235).

For immunoblotting, 5 µg of protein lysate (mixed *v*/*v* with 1X loading buffer (4% *w*/*v* SDS, 20% *v*/*v* glycerol, and 0.01% *v*/*v* bromophenol blue, 0.125 mM Tris-hydrochloride (Tris-HCl), pH 6.8) with 1M dithiothreitol (DTT)) was separated by electrophoresis on a NuPAGE™ 4–12% Bis-Tris gel (Invitrogen, NP0321) for 1 h, 150 V. Proteins were transferred onto nitrocellulose membrane using the iBlot™ 2.0 Dry Blotting System (20 V, 14 min; Life Technologies, Carlsbad, CA, USA). Membranes were blocked using skim milk powder in Tween PBS (1 × PBS, 0.1% (*w*/*v*) Tween 20) (TPBS) for 1 h while shaking at RT, and rinsed three times with TPBS for 5 min. Membranes were incubated with primary mouse or rabbit antibodies against α-SMA 1:1000 dilution (Abcam, ab5694) and GAPDH 1:1000 dilution (Cell Signaling Technology, Danvers, MA, USA D4C6R/97166S), in TPBS, overnight at 4 °C. Membranes were washed with TPBS and incubated with secondary antibodies (1:20,000); IRDye 800CW goat anti-mouse antibody or IRDye 680RD goat anti-rabbit antibody (LI-COR Biosciences, Lincoln, NE, USA, 926-68071 and 926-32210), for 1 h while shaking at RT. Membranes were washed 3 times with TPBS and imaged using Odyssey Infrared Imaging System (LI-COR Biosciences, Lincoln, NE, USA), at 700 nm and 800 nm.

### 4.11. Proteomics: Solid-Phase-Enhanced Sample Preparation

iPSCs and generated NVs (*n* = 4), and target cells following NV treatment (CMs, HUVECS, hCFs) (*n* = 4) were lysed in 1% *v*/*v* sodium dodecyl sulphate (SDS), 50 mM triethylammonium bicarbonate (TEAB), pH 8.0, incubated at 95 °C for 5 min and quantified by microBCA (Thermo Fisher Scientific) as described [143]. Proteomic sample preparation was performed from 10 µg of protein extract as previously described [142] using single-pot solid-phase-enhanced sample preparation (SP3) [144]. Briefly, samples were reduced with 10 mM DTT at RT for 1 h (350 rpm), alkylated with 20 mM iodoacetamide (IAA) (Sigma-Aldrich) for 20 min at RT (light protected), and immediately quenched with 10 mM DTT. A Sera-Mag SpeedBead carboxylate-modified magnetic particle mixture (hydrophobic and hydrophobic 1:1 mix, 65152105050250, 45152105050250, Cytiva) were added to each protein extract, washed in 50% ethanol, and incubated for 10 min (1000 rpm) at RT. Beads were sedimented on a magnetic rack, supernatants removed and beads washed three times with 200 µL 80% ethanol. Beads were resuspended in 100 µL 50 mM TEAB pH 8.0 and digested overnight with trypsin (1:50 trypsin: protein ratio; Promega, V5111) at 37 °C, 1000 rpm. The peptide and bead mixture was centrifuged at 20,000× g for 1 min at RT. Samples were placed on a magnetic rack and supernatant was collected and acidified to a final concentration of 1.5% formic acid (FA), frozen at -80 °C for 20 min, and dried by vacuum centrifugation for ~1 h. Peptides were resuspended in 0.07% trifluoroacetic acid (TFA), quantified by Fluorometric Peptide Assay (Thermo Fisher Scientific, 23290) as per manufacturer’s instructions, and samples normalised with 0.07% TFA.

### 4.12. Proteomics: Nano Liquid Chromatography–Tandem Mass Spectrometry

Peptides were analysed on a Dionex UltiMate NCS-3500RS nanoUHPLC coupled to a Q-Exactive HF-X hybrid quadrupole-Orbitrap mass spectrometer equipped with nanospray ion source in positive, data-dependent acquisition mode [12,145]. Peptides were separated using high resolution analytical nano liquid chromatography separated (1.9-µm particle size C18, 0.075 × 250 mm, Nikkyo Technos Co., Ltd., Tokyo, Japan) with a gradient of 2–28% acetonitrile containing 0.1% formic acid over 95 min at 300 nL min^−1^ followed by 28–80% from 95–98 min at 300 nL min^−1^ at 55 °C (butterfly portfolio heater, Phoenix S&T). An MS1 scan was acquired from 350–1650 *m*/*z* (60,000 resolution, 3 × 10^6^ automatic gain control (AGC), 128 msec injection time) followed by MS/MS data-dependent acquisition (top 25) with collision-induced dissociation and detection in the ion trap (30,000 resolution, 1 × 10^5^ AGC, 60 msec injection time, 28% normalized collision energy, 1.3 *m*/*z* quadrupole isolation width). Unassigned, 1, 6–8 precursor ions charge states were rejected and peptide match disabled. Selected sequenced ions were dynamically excluded for 30 s. Data was acquired using Xcalibur software v4.0 (ThermoFisher Scientific). The MS-based proteomics data and analysis parameters have been deposited to the ProteomeXchange Consortium via the PRIDE partner repository with the dataset identifier PXD036654.

Replicates: cell proteome (iPSCs/HUVECs) *n* = 4; NVs (CL2 and CERA) *n* = 4; NV-treated cell proteome (CMs, HUVECs, hCFs) *n* = 4.

### 4.13. Proteomics: Data Processing and Informatics/Visualisation

Identification and quantification of peptides was performed using MaxQuant (v1.6.14.0) with its built-in search engine Andromeda [146], as described [12,143,147]. Human-only (UniProt #78,120 entries) sequence database (March 2021) with a contaminants database was employed. N-terminal acetylation and methionine oxidations were set as variable modifications. False discovery rate (FDR) was 0.01 for protein and peptide levels. Enzyme specificity was set as C-terminal to arginine and lysine using trypsin protease, and a maximum of two missed cleavages allowed. Peptides were identified with an initial precursor mass deviation of up to 7 ppm and a fragment mass deviation of 20 ppm. Protein identification required at least one unique or razor peptide per protein group. Contaminants, and reverse identification were excluded from further data analysis. ‘Match between run algorithm’ in MaxQuant [148] and label-free protein quantitation (maxLFQ) was performed. All proteins and peptides matching to the reversed database were filtered out.

Perseus [149] and R studio were used to analyse the proteomic data and generate plots. G:Profiler, Reactome, STRING, and Cytoscape (v3.9.1) were used for enrichment analysis. Protein lists for samples were generated in Perseus (v1.6.14.0) [150]. For cell, NV, and NV-treated cell proteomes, proteins were identified at least once in two biological replicates per group. Protein intensities (maxLFQ) were log_2_ transformed. Statistical analysis and plots were generated using Perseus, R studio and GraphPad Prism. Principal component analysis, Pearson correlation matrix, and hierarchical clustering was performed in R studio using Euclidian distance and average linkage clustering, with missing values imputed at z-score 0 for heatmap generation only. R was used for data visualisation (ggplot2, ggpubr packages). In all instances significance was *p* < 0.05 unless otherwise indicated.

## Figures and Tables

**Figure 2 ijms-23-14334-f002:**
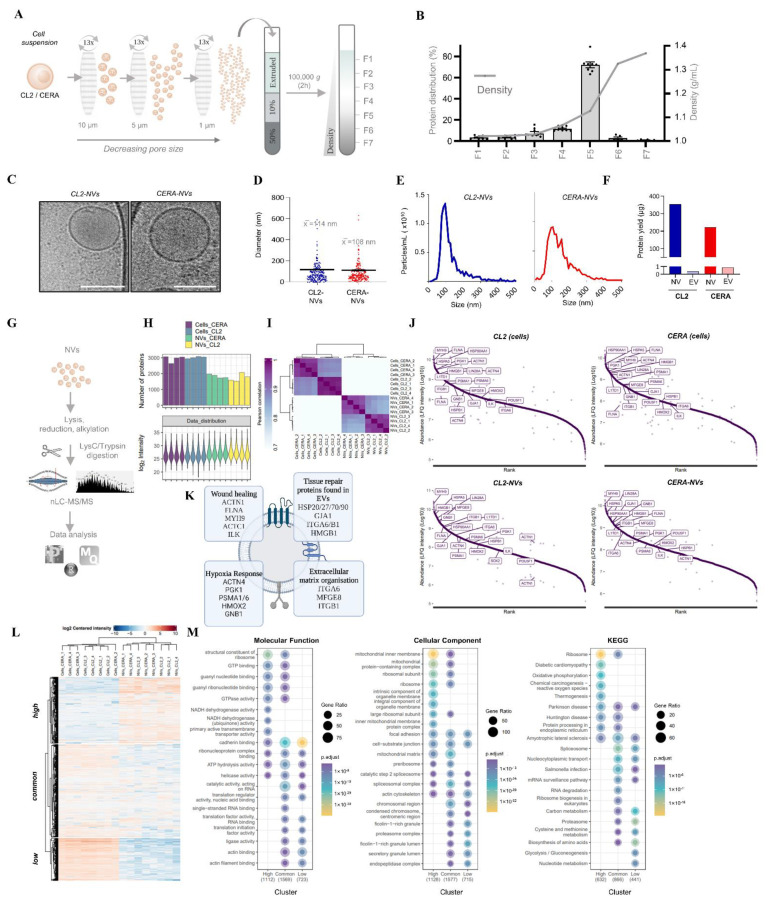
Generation and characterisation of NVs generated from human-induced pluripotent stem cells. (**A**) iPSC nanovesicle generation using serial extrusion (10, 5, 1 µm filter, 13× each membrane) with the cell suspension purified using density-cushion ultracentrifugation to obtain fractions (F) 1–7 of increasing density. (**B**) The buoyant densities of seven fractions collected for each iPSC preparation were determined by absorbance at 244 nm using a molar extinction coefficient of 320 L g^−1^cm^−1^. Protein yield determined based on micro-BCA protein quantification (as a % of total), revealing major fraction F5 as NV-containing. Data presented as mean ± s.e.m. (**C**) Cryo-electron microscopic analysis of CL2- and CERA-derived NVs (F5), scale 100 nm. (**D**) Scatter plot distribution of NV particle diameter as determined by cryo-EM images. Data presented as mean ± s.e.m (standard error of mean). (**E**) Size distribution profiles of NVs determined by single particle tracking analysis (ZetaView), indicating mean ~110 nm. (**F**) Protein yields (determined by microBCA) of NVs (F5, density-based preparation) and natural EVs (EVs, density-based preparation) from CL2 and CERA iPSCs. Yields correspond to the same starting cell number for each cell model. (**G**) Workflow for mass spectrometry-based proteomic profiling of NVs. (**H**) Protein identification and LFQ (normalised) intensity for NVs and cell lysates from each iPSC model (*n* = 4). (**I**) Pearson correlation (R-score) of NVs and iPSCs (*n* = 4). (**J**) Abundance distribution (waterfall plots, LFQ intensity, log_10_) of proteins identified in iPSCs (cells) and derived NVs. Several examples of proteins similarly identified between both cells and NVs are shown. (**K**) Selected proteins identified in NVs associated with wound healing, hypoxia response, extracellular matrix organisation and implicated in tissue repair found in natural EVs. (**L**) Hierarchical cluster analysis of iPSCs and NVs (*p* < 0.05, fold change (FC): 1.5), reveals three clusters of differential protein expression; *high* abundant NVs, *low* abundant NVs, similarly expressed (*common*) between NVs and iPSCs. (**M**) Gene Ontology enrichment analysis (ranked, *p* < 0.05) of each cluster based on cellular component, biological processes, and Reactome pathway analyses.

**Figure 3 ijms-23-14334-f003:**
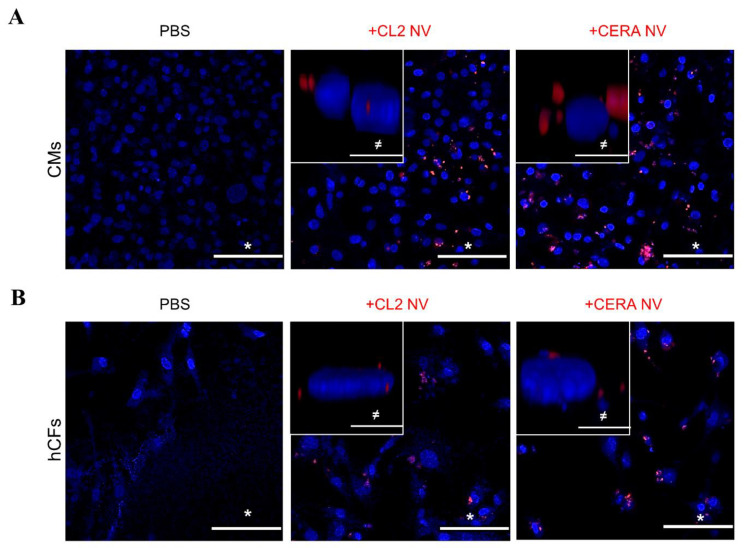
Internalisation of NVs by cardiac fibroblasts and cardiomyocytes. Uptake of iPSC-derived NVs by (**A**) human cardiomyocytes (CMs) and (**B**) human primary cardiac fibroblasts (hCFs). Fluorescence microscopy analysis of cells incubated with NVs labelled with lipophilic dye (DiI) for 2 h. DiI vehicle (PBS) control. Nuclei (blue) were stained with Hoechst. Scale bar, * 100 μm, ≠ 10 μm.

**Figure 4 ijms-23-14334-f004:**
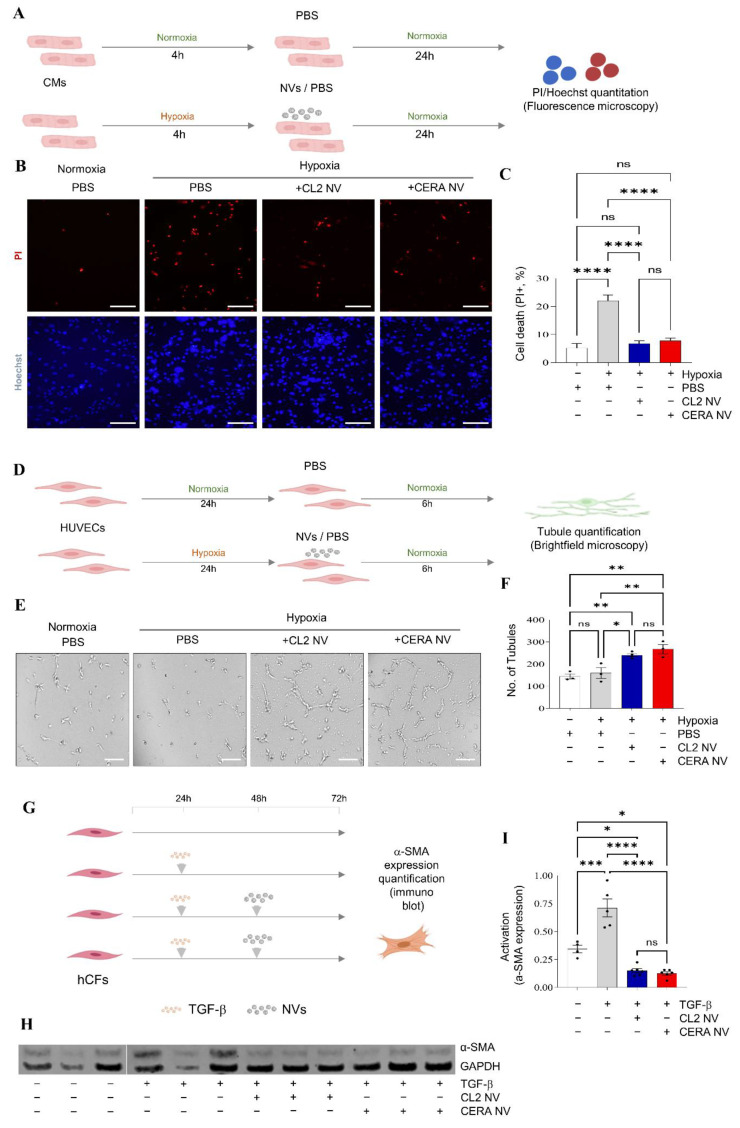
iPSC NVs functionally regulate aspects of cardiac repair in vitro. (**A**) Cardiomyocyte (CM) survival assay, where cells are exposed to hypoxia/normoxia (4 h) and treated with NVs under normoxic condition for 24 h followed by fluorescence microscopy. (**B**) NV single dose treatment (6 µg in 200 µL) confers reduced CM cell death based on % propidium iodide (PI) stain, Scale bar, 500 μm. (**C**) Bar plot for cell survival shown as percentage of cell death for NV treatments (CL and CERA) and vehicle and untreated controls (*p* < 0.0001). (*n* = 2 biological, 3 technical). Data represented as mean ± s.e.m. (**D**) Endothelial (HUVEC) tube formation assay in response to NV treatment following hypoxia/reoxygenation. (**E**) Brightfield microscopy images of tube formation assay (normoxic and hypoxic) in response to NV treatment (single dose, 1.5 µg in 50 µL), Scale bar, 200 μm. (**F**) Histogram analysis of the number of tubules formed for NV treatments (CL and CERA) and vehicle and untreated controls (*p* < 0.05 and <0.005, respectively). (*n* = 2 biological, 5 technical), Data represented as mean ± s.e.m. (**G**) TGF-β-mediated (5 ng/mL) human primary cardiac fibroblast (hCF) activation (24 h) or PBS vehicle; smooth muscle actin activation (α-SMA) assessed at 72 h. (**H**) NV single dose treatment (24 h post TGF-β stimuli); 9 µg in 300 µL confers reduced α-SMA expression based on immunoblot analysis compared to TGF-β alone and GAPDH. (**I**) Quantitation of differential α-SMA expression (*n* = 2 biological, 3 technical). Data represented as mean ± s.e.m. (*p* < 0.0001). * *p* < 0.05, ** *p* < 0.005, *** *p* < 0.0005, **** *p* < 0.0001.

**Figure 5 ijms-23-14334-f005:**
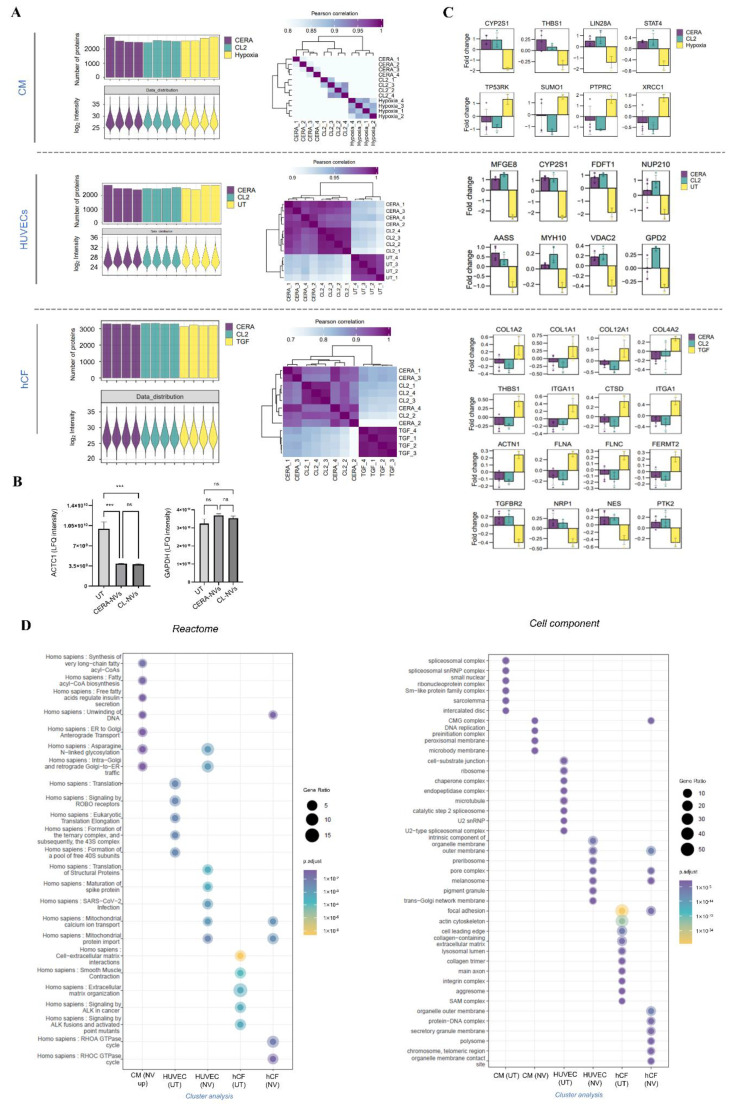
iPSC NVs induce target cell proteome reprogramming to support repair functions. (**A**) Cell proteome analysis of cell assays following NV treatment: cardiomyocyte (CM) survival, endothelial (HUVEC) tube formation, and TGF-β-mediated human primary cardiac fibroblast (hCF) activation. Protein identification, LFQ (normalised) intensity, and Pearson correlation for cells in response to NV treatment, and vehicle control for each assay (*n* = 4). (**B**) MS-based quantitation (LFQ intensity) of ACTC1 (actin alpha cardiac muscle 1) or GAPDH expression in hCFs in response to TGF-β (vehicle, UT) and following NV treatment (CERA- or CL2-NVs). Data represented as mean ± s.e.m. *** *p* < 0.0005, ns, non-significant. (**C**) Differential protein abundance (fold change, of LFQ intensity, log_2_) of selected protein markers for each assay following NV treatment (CERA or CL2), relative to vehicle controls (yellow) (*p* < 0.05). Proteins selected include pro-survival, pro-angiogenic, and anti-fibrotic associated markers identified in study. (**D**) Gene Ontology enrichment analysis (ranked, *p* < 0.05) (Reactome and GO cellular component) of proteins identified differentially enriched for each assay. Hierarchical cluster analysis of NV treatments and vehicle controls for each assay were determined (Appendix A, ANOVA, *p* < 0.05, fold change (FC): 1.5), to reveal distinct clusters of differential protein expression for each assay. Differential protein subsets for each assay (Appendix A) were then mapped using Reactome and GO cellular component and plotted as adj. *p*-val for each category/assay/treatment.

## Data Availability

Data generated or analysed during this study are included in this published article (and its Appendix A files) or available from Data Repositories. Raw proteome data (cell, NV, and NV-treated cell proteomes) and parameter/search information is available from the ProteomeXchange Consortium via the PRIDE partner repository (#PXD036654, http://www.proteomexchange.org/; accessed 12 September 2022). Functional enrichment annotations were retrieved using g:Profiler (https://biit.cs.ut.ee/gprofiler/; accessed 12 September 2022). Hierarchical clustering was performed in R and Perseus using Euclidian distance and average linkage clustering, with missing values imputed at z-score 0. R was also used for data visualisation.

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
