# Peer review of "Scalable Generation of Nanovesicles from Human-Induced Pluripotent Stem Cells for Cardiac Repair"

_ijms, 2022, doi:10.3390/ijms232214334_

Round 1

Reviewer 1 Report

In this study, Lozano et al. reported a scalable generation EV-like membranous nanovesicles (NVs) from human iPSCs, able to be produced in large quantities of to be intact and functional. NVs were isolated using density-gradient separation avoiding as reported the presence of contaminants and were able to be internalized by human cardiomyocytes, primary cardiac fibroblasts, and endothelial cells.

NVs captured the dynamic proteome of parental cells including pluripotency markers, tissue repair, and extracellular matrix organization proteins. Moreover, NVs after internalization were able to block apoptosis stimulate angiogenesis, and attenuate extracellular matrix remodeling in different cell types. The paper provides a scalable approach for the generation of functional NVs for clinical application for tissue repair.

Major comments

-           NVs isolation was performed by density-gradient separation, supporting a clear elimination of the possible contaminants in the preparation. Despite that, the absence of protein signal (shown in the graph of Figure 2B) in almost all the other fractions, let us question if all the cellular proteins were encapsulated in the NVs. The authors should also measure EV concentration also by NTA in all the fractions isolated. The NTA data should be performed also to compare NVs and naïve EVs in Figure 2F. Moreover, electron microscopy evaluation should be performed also for the other fractions to confirm their selective presence in Fraction 5.

- The NVs were compared with naïve EVs only for their concentration. The comparison should be also performed in the uptake and functional experiments of Figures 3 and 4. Are the NVs mimicking the same activity of naïve EVs or are they more potent or different with respect to naïve EVs?

- The survival assay was performed by PI labeling. This labeling is not sufficiently precise as functional assay to confirm the protection for apoptosis. The authors should add another assay such as Annexin/PI staining for apoptosis or BrdU staining for proliferation evaluation.

- The fibrotic data are unclear. The authors demonstrated no significant activity of NVs with respect to vehicle alone as reported in Figure 4G-H on cardiac fibroblasts. The authors should add the EVs during TGF-β treatment to check their negative regulation of extracellular matrix remodeling induced by this stimulation in cardiac fibroblasts. The same can be proposed for the MS-based data, in Figure 5, where the correct comparison should be between vehicle and NVs treatment alone, or between the presence or absence of NVs during the TGF-β treatment.

Minor comments

The authors should add in the introduction a paragraph describing the regenerative properties of naïve iPSC-EVs in different organs (such as the heart and kidney). The papers already published demonstrated this activity should also be included in the text.

Author Response

Please see detailed response to each reviewer query attached

Reviewer 2 Report

In this manuscript, the authors took advantage of a previously reported method of nanovesicle purification from cell lysates to demonstrate that a size-based extrusion method to purify iPSC nanovesicles reach a much higher yield than the traditional side exclusion chromatography or ultracentrifugation of extracellular vesicles from culture medium. More importantly, they demonstrate at a proteomic and functional levels, that EVs and nanovesicles are equivalent, which is important from a translational point of view. In the opinion of this reviewer, the methodology is well explained, the experiments are well performed and overall, this paper would be of interest for researchers interested in the isolation of large amounts of  clinical-grade nanovesicles for regenerative therapies.

Author Response

please see response to reviewer comment attached

Round 2

Reviewer 1 Report

In the revised paper, the authors explain important elements that were lacking in the initial paper. The extrusion process was better clarified to understand in detail how it functions and how can be improved, as well as a clear comparison between EVs and NVs on their intrinsic properties was added, when possible. The references were added accordingly and give more background to the study.

A clarification of the fibrotic experiment was also provided. I would suggest a small modification to the illustration in Figure 4G. The authors should add the name CL2 and CERA NVs, to distinguish the two images that compose panel G (with NVs).